## RESEARCH ARTICLE

# Physical confinement and phagocytic uptake induce persistent cell migration

Summer G. Paulson[1,2], Sophia Liu[1,2] and Jeremy D. Rotty[1,*]

## ABSTRACT

Physical confinement is not routinely considered as a factor that influences phagocytosis, which is typically investigated using unconfined *in vitro* assays. BV2 microglia-like cells were used to interrogate the impact of confinement on IgG-mediated phagocytosis side by side with unconfined cells. Confinement acted as a potent phagocytic driver, greatly increasing the fraction of phagocytic cells in the population compared to the unconfined setting. Arp2/3 complex and myosin II contributed to this effect. Remarkably, confinement partially rescued phagocytic uptake upon myosin II disruption. In addition, cells under confinement were partially resistant to the actin-depolymerizing drug cytochalasin D. Unexpectedly, we observed that bead uptake stimulated persistent migration, a process we term 'phagocytic priming'. Integrin-dependent adhesion was required for phagocytic priming in unconfined and confined settings but was dispensable for phagocytic uptake. The cytoskeletal requirements for phagocytic priming differed depending on confinement state. Myosin II and Arp2/3 complex were required for phagocytic priming under confinement, but not in unconfined settings. As with phagocytosis, cytoskeleton-dependent priming of motility varies based on physical confinement status. Phagocytic priming may be a crucial innate immune mechanism by which cells respond to wounds or trauma with increased surveillance of the local microenvironment.

KEY WORDS: Microglia, Confined motility, Phagocytosis, Actin, Myosin, Arp2/3 complex, Fibronectin

## INTRODUCTION

Many microenvironmental factors act as external stimuli to influence cell morphology or function (Paul et al., 2017; Truffi et al., 2020). These include chemical cues like chemotactic or haptotactic gradients that elicit directed cell migration, as well as physical cues like tissue stiffness and physical confinement that alter cell shape or modify cellular responses. Confinement, defined as any external cue that restricts cell morphology or mobility (Ilina et al., 2020; Wong et al., 2018), is another factor that can synergize with other chemical and physical cues. These and other microenvironmental stimuli crucially influence the cell's response to its environment and surrounding cells, including by prompting a phagocytic response or altering motility (Paul et al., 2017; Truffi et al., 2020).

The physical microenvironment alone is sufficient to alter cellular responses. For example, cells reintroduced into decellularized extracellular matrix (ECM) structures assume identities similar to the missing cell population (Capella-Monsonís et al., 2023). Hydrogels derived from decellularized ventricular ECM increased the number of endogenous cardiomyocytes in rats (Singelyn et al., 2012) and decellularized, demineralized bone matrices bonded with vascular endothelial growth factor increased endothelial cell proliferation and promoted angiogenesis via new microvessel invasion within the scaffolds (Chen et al., 2010). In microglia, persistent migration can also be stimulated by the physical microenvironment through mechanical stretch (Procès et al., 2024). In other cell types, persistent migration has been induced by confinement (Irimia and Toner, 2009; Petrie et al., 2009; Rolli et al., 2010; Stinson et al., 2025).

Despite these insights, the microenvironment's influence on microglia function remains understudied in comparison. Microglia, the resident immune cell of the central nervous system (CNS), are responsible for maintaining CNS homeostasis by supporting synapse formation, clearing debris, and responding to inflammatory cues within the brain and spinal cord (Brabazon et al., 2018; Kim et al., 2013; Melo et al., 2022; Parkhurst et al., 2013). The brain itself is a highly confined space (Fig. 1A) (Soles et al., 2023), highlighting the importance of considering confinement when assaying brain-derived cell types *in vitro*. This need is far from hypothetical. For instance, standard unconfined *in vitro* microglia cultures exhibit amoeboid-like morphologies, which starkly deviate from the well-documented ramified morphology of *in vivo* microglia (Jeong et al., 2013; Karperien et al., 2013). Amoeboid cells are more migratory and active than ramified ones, and therefore more likely to phagocytose, meaning that this morphological divergence also likely retains functional significance (Vidal-Itriago et al., 2022). One explanation for the divergence is that *in vitro* assays lack crucial microenvironmental cues characteristic of the *in vivo* environment, including confinement. However, it is especially difficult to examine the effects of confinement and other microenvironmental factors on cellular behavior in the brain, where decellularizing and reinjecting ECM matrix is less feasible than in other tissues. A recent study by Sharaf et al. assayed microglial growth in 'two and a half dimensions' (2.5D) by placing them across CAD-printed nano-pillar arrays, with '2.5D' indicating that the nano-pillars provide increased topology and dimensionality compared to a flat surface but are distinct from a true three-dimensional (3D) assay (Sharaf et al., 2022). Remarkably, this strategy recovered the ramified cell morphology compared to amoeboid-like cells in standard two-dimensional (2D) culture (Sharaf et al., 2022). However, the study used relatively hard surfaces for their pillars, reporting a Young's modulus of 0.25-11.4 MPa. The Young's modulus of brain tissue is significantly

[1]Uniformed Services University of the Health Sciences, Department of Biochemistry, Bethesda, MD, USA, 20814. [2]The Henry M. Jackson Foundation for the advancement of Military Medicine, Bethesda, MD, USA, 20817.

*Author for correspondence (Jeremy.Rotty@usuhs.edu)

S.G.P., 0000-0002-3488-1817; J.D.R., 0000-0002-7782-9672

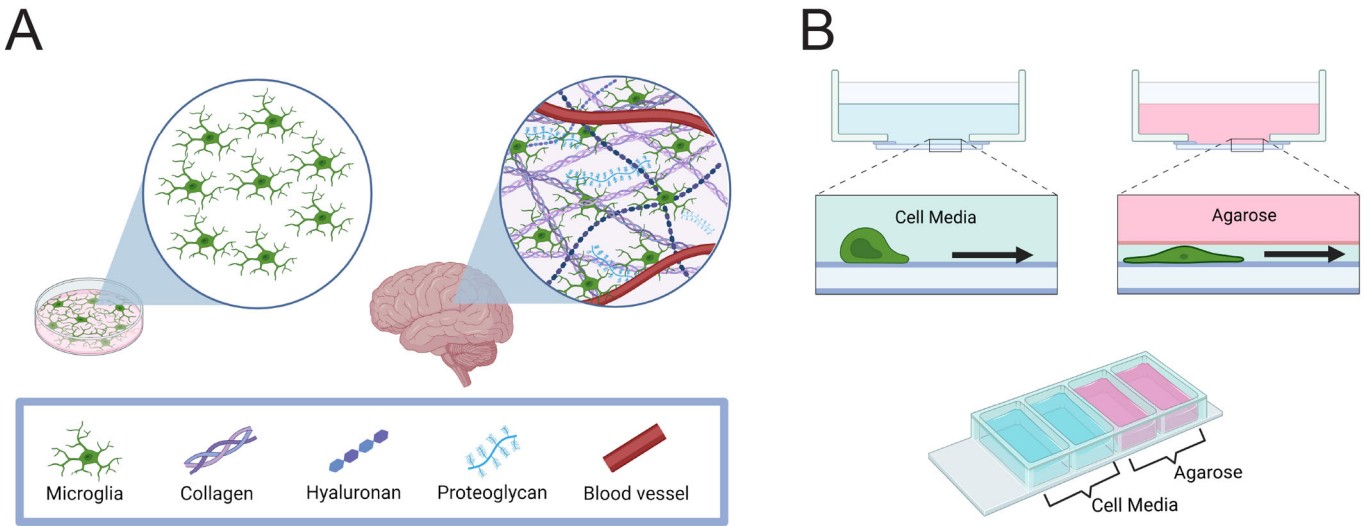

**Fig. 1. The brain is a confined environment.** (A) Schematic demonstrating microenvironmental components confining cells in the brain, compared to cells in a Petri dish that are allowed to move freely. Created in BioRender by Paulson, S. (2025) https://BioRender.com/btwt9d3. (B) Top: examples of cells moving with and without a confining layer of agarose. Bottom: schematic of cell assay used throughout. A four-chamber glass-bottomed dish was coated with 10 µg/mL fibronectin and then filled with either complete cell media or 1% agarose. Cells were introduced into wells and allowed to move. Created in BioRender by Paulson, S. (2025) https://BioRender.com/khn4ts9.

softer, ranging from 0.1-1 kPa (Iwashita et al., 2014; Lu et al., 2006). In addition, CAD-printed nano-pillars are relatively challenging to generate, and do not incorporate physical confinement as an experimental parameter. A common confinement strategy used in chemotaxis studies confines cells under 1% agarose, which has a stiffness in the same range as brain (Heit and Kubes, 2003; Mori et al., 2013).

When seeking to model the physiological environment of the brain, there are many other factors to consider. Elastic, soft confining environments composed of agarose, collagen, or Matrigel have all been used to interrogate the role of the actin cytoskeleton in cellular behavior. A study performed on dendritic cells under agarose demonstrated that increased agarose stiffness resulted in decreased cell velocity and distance travelled during cell migration (Gaertner et al., 2022). With heightened agarose stiffness, the cells created actin patches, using WASp and the Arp2/3 complex to generate actin patches to push against the plasma membrane and raise the confining environment upward so that the cell could propel itself forwards. A study with cells plated on polyacrylamide gels demonstrated that the introduction of fibronectin micropatterns resulted in increased adhesion and prevented cell spreading, instead confining the cells to the fibronectin micropatterns (Yousafzai et al., 2024). They found that cells in these fibronectin patches on the polyacrylamide gel had an altered flow of F-actin within the cells, moving in a retrograde fashion instead of outward to contain the cells to the adhesion patch. Confining macrophages with either a micropattern or by cell crowding downregulated the pro-inflammatory cytokine secretion that precedes phagocytosis (Jain and Vogel, 2018). Another study using bone-marrow-derived macrophages in Matrigel found that inhibition of F-actin, Rho-associated coiled-coil containing kinases (ROCK), or non-muscle myosin II (NMII) decreased cell migration speed compared to untreated confined cells (Paterson and Lämmermann, 2022). Each of these results points to the importance of several components of the actin cytoskeleton in confined migration. However, limited research exists about how phagocytosis occurs in confined environments similar to most *in vivo* settings.

Therefore, we set out to examine the effects of confinement on microglia phagocytosis by directly comparing IgG phagocytosis in confined and unconfined cells, using agarose with a stiffness akin to brain parenchyma. We discovered that confinement promotes more efficient phagocytosis, an effect that is dependent upon Arp2/3 complex. In contrast, we noted that confinement continues to promote phagocytosis after inhibition of myosin II or inhibition of new filament formation via cytochalasin D treatment compared to analogous treatments on unconfined cells. These findings imply that there may be an actomyosin-independent phagocytic component that remains activated under physical confinement. We also discovered that confinement and phagocytic uptake work together to induce 'phagocytic priming', where cells migrate in a highly persistent fashion immediately after phagocytosis. Phagocytic priming required integrin-dependent adhesion to the ECM in confined and unconfined settings, while Arp2/3 complex and myosin II function were only required under confinement. As a whole, this work supports the idea that the molecular requirements for phagocytosis and cytoskeleton-dependent priming may be context-dependent and especially responsive to physical confinement. In addition, phagocytic priming may be crucial for supporting innate immune processes like antigen-presenting cells homing to the lymph node, enhanced surveillance of the microenvironment, or phagocytic cells responding to wounds and infection.

## RESULTS
### Confinement enhances phagocytic uptake and persistent migration

Many of the molecular mechanisms pertaining to glial cell biology have been discovered and interrogated chiefly in 2D culture systems. Therefore, we designed a low-cost, accessible assay to interrogate the influence of confinement on cell motility and phagocytosis. Using a four-chamber dish, cells were either left unconfined in a well filled with media or confined under a layer of 1% agarose (Fig. 1B, bottom). These assays are based on previous cell motility experiments in the lab (Stinson et al., 2025), in turn inspired by the classic under-agarose chemotaxis assay as described

by Heit and Kubes (2003). This setup allowed us to examine cell function in response to physiologically relevant cues under confinement (Fig. 1B, top).

First, we compared the ability of microglia-like BV2 cells to phagocytose unconfined in media to their phagocytic ability when confined under 1% agarose. Murine IgG was labeled with a pHrodo fluorescent tag and used to opsonize 2 μm polysphere beads. The pHrodo tag was used because it fluoresces under acidic environments, meaning that uptake by the acidic phagolysosome would result in bead fluorescence. Thus, we were able to detect complete phagocytosis in an unbiased fashion over 8-h time-lapse movies (Fig. S1A). To rule out any signal from external beads, we compared external and internalized beads and confirmed that the two states are clearly distinguishable (Fig. S1B). To normalize the number of beads present when comparing between media and confined wells, bead densities were measured and categorized into different density groups (Fig. S1C,D). Beads injected under agarose did not disperse as evenly as in media wells. Due to this variability between fields of view under confinement, the number of beads was calculated, allowing comparisons between confined and media wells to be controlled for bead density. Because the majority of unconfined fields contained 'low bead density' (Fig. S1D, bottom), analysis was conducted exclusively with 'low bead density' fields of view for both unconfined and confined conditions (Fig. S1D, middle). We also confirmed that washing in beads did not cause cells to become unconfined long-term under agarose. Fluorescent dextran was polymerized into the agarose, and then BV2 cells were subjected as normal to pHrodo-IgG bead uptake. Representative images demonstrate that the dextran signal wraps around cells, and that some areas of the cell push against the agarose to raise the confinement ceiling (Fig. S2A shows two example fields of view, denoted i and ii). Furthermore, BV2 cells under agarose were compressed to roughly half the height of unconfined cells as they began to internalize beads (Fig. S2B). These data demonstrate that phagocytic BV2 cells are confined when they encounter beads washed underneath the agarose.

Phagocytosis was measured at the 2-h mark, as cells after this time point became saturated with beads and stopped phagocytosing. Example images for both the media well (Fig. 2A) and the confined well (Fig. 2B) demonstrate pHrodo fluorescence signal from internalized beads. Cells under confinement were significantly more phagocytic than unconfined (media) cells (Fig. 2C; Fig. S2C). While more cells became phagocytic under confinement, there was no overall difference in phagosome size or number when comparing pHrodo+ cells in either condition (Fig. 2D-F). Together, these data demonstrate that confinement induces a higher number of cells to phagocytize but does not affect the efficiency of intracellular trafficking to the phagosome after bead uptake.

From the time lapse movies, it became apparent that phagocytosis altered the migratory trajectory of cells. Specifically, bead uptake seemed to induce more persistent migration tracks compared to non-phagocytic cells (Movie 1). Therefore, we decided to investigate whether bead uptake primed cells to move more persistently. We measured cell velocity, accumulated migration distance, and migratory persistence. Persistence (D/t) is a measurement of a cell's track from start to finish (t) and its net displacement during that timeframe (D), with values closer to 1 indicating more persistent migration. While directionality describes the tendency of a cell to migrate in one direction toward a particular cue (e.g. chemotaxis or haptotaxis), persistence refers to a cell's ability to maintain its migratory bearing even in the absence of a directional cue. Increasing confinement pressure by placing cells under higher-

percentage agarose gels led to decreasing velocity and accumulated distance, but no coherent persistence trend (Fig. S3A-C). With these data in hand, we returned to the setup from Fig. 1C, except that each condition (media and confined) had one well with cells alone and one well with cells plus beads. We tracked cells for 8 h and compared the cell velocity, accumulated distance, and cell persistence between groups. Notably, the −IgG beads condition revealed only a confinement-induced decrease in migration velocity (Fig. 2G-I). When considering the +IgG beads as a bulk population (i.e. not separating out post-phagocytosis cells), we likewise saw no significant confinement-induced migration trends (Fig. 2G-I). However, substantial differences were revealed when we narrowed our analysis to the post-phagocytic phase. Confined and unconfined post-phagocytic cells migrated slowly, but with higher persistence than non-phagocytic cells (compare −IgG populations to post-phagocytic populations, Fig. 2G-I). Furthermore, confined post-phagocytic cells were significantly more persistent, but slower, than even unconfined post-phagocytic cells (Fig. 2G-I). Qualitatively, this can be seen from investigating the individual cell paths from confined and unconfined settings ±bead uptake. The paths are longer and straighter in general under confinement compared to unconfined cells (Fig. 2J; example tracks in Movie 2). In addition, phagocytic uptake renders cells more likely to pursue a straighter path in each setting compared to the condition-matched 'all tracks' readout (Fig. 2J). Of all the conditions, the confined post-phagocytosis cells maintained the longest, straightest cell tracks (Fig. 2J; Movie 2), in line with the previously mentioned quantitative measures. Henceforth, unless otherwise indicated, we will discuss cell migration measurements only in post-phagocytic populations. These data point towards a process we have termed 'phagocytic priming', where phagocytosis stimulates persistent migration. Confinement further increases persistent migration, enhancing the phagocytic priming response. Our next goal was to determine the cytoskeletal regulators involved in phagocytic priming and phagocytosis under confinement.

## The Arp2/3 complex is required for phagocytic priming and uptake

We first interrogated the Arp2/3 complex, a seven-subunit branched actin nucleator responsible for generating the branched actin networks that form lamellipodial protrusions linked to environmental sensing (Goley and Welch, 2006). While the Arp2/3 complex has been implicated in phagocytosis in unconfined settings (Rotty et al., 2017), there is less known about its function during confined phagocytosis. Since the Arp2/3 complex is responsible for creating lamellipodial structures at the leading edge of polarized cells (Tokuraku et al., 2020), we hypothesized that the Arp2/3 complex would be integral to both phagocytic uptake and priming under confinement.

We reused our four-chambered dish, and this time added the Arp2/3 complex small molecule inhibitor CK-666 or vehicle control (DMSO) into the media or the agarose gel (Fig. S3D). Trial runs were performed comparing cells with and without DMSO treatment to determine whether vehicle treatment altered normal cellular behavior (Fig. S3E-K). DMSO had no significant impact on phagocytosis under confinement, but did slightly raise phagocytic cell number and decrease phagosome number per cell in media (Fig. S3E-H). With respect to motility, DMSO did not significantly affect velocity, distance traveled, or persistence under confinement or in media (Fig. S3I-K). We concluded based on these findings that DMSO is an appropriate vehicle control for our inhibitor treatment studies. Any significant decrease between DMSO vehicle and drug-

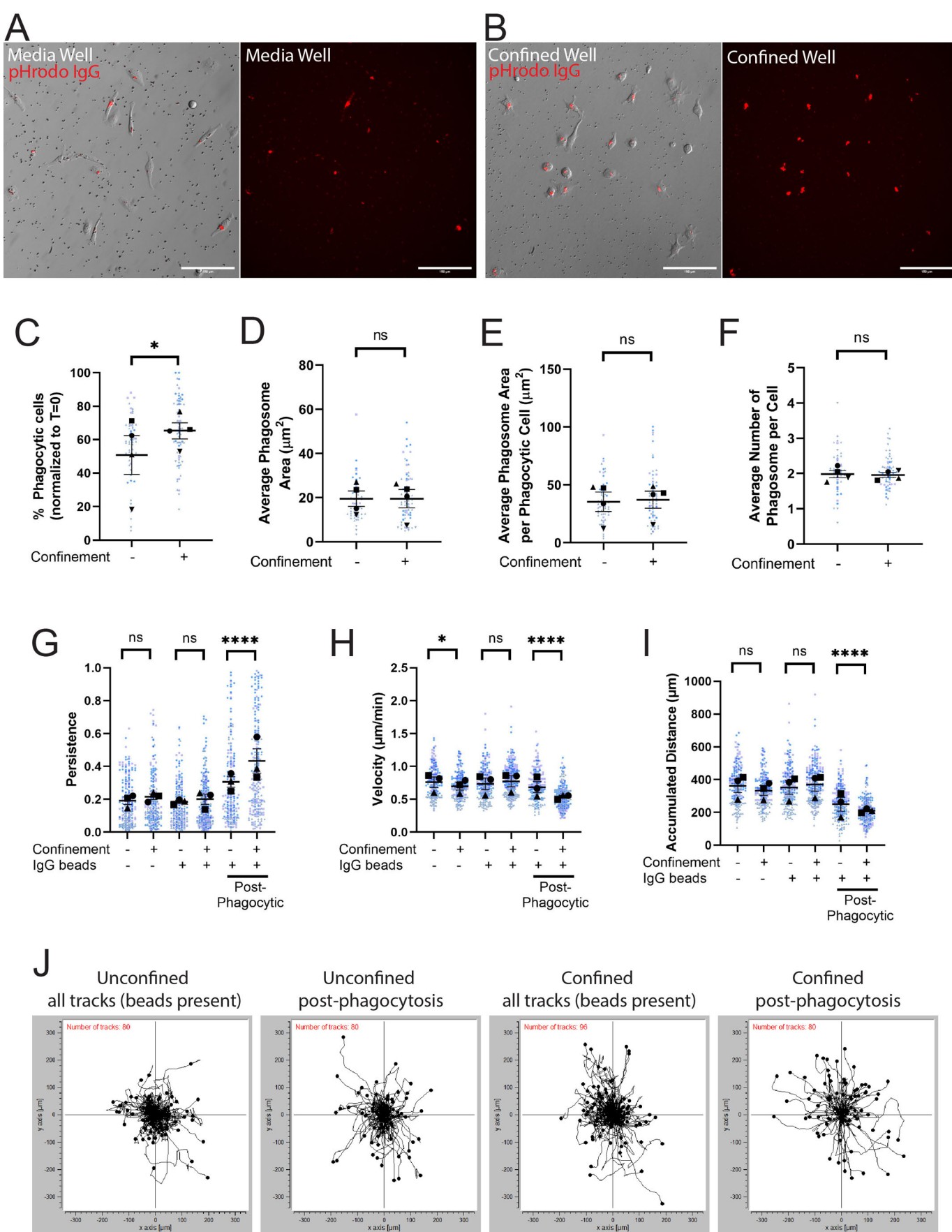

**Fig. 2.** See next page for legend.

**Fig. 2. Confinement enhances phagocytosis and primes post-phagocytosis cells to migrate persistently.** (A,B) Example composite images of phase contrast and pHrodo-red merged images 2 h after pHrodo-IgG beads were added to media wells (A) or confined agarose wells (B). The pHrodo-red signal from these merges has been isolated alongside the merge to emphasize internalized pHrodo-IgG-bead signal. Scale bar: 100 µm. (C) The percentage of fluorescent cells in a field of view, comparing unconfined and confined cells, normalized to T=0. Any cell fluorescent at T=0 was removed from further counting. (D) Average phagosome size ($µm^2$). This is the average size of phagosomes per field of view divided by the average number of phagosomes per field of view. (E) Average phagosome area per phagocytic cell ($µm^2$). This was calculated by dividing the average size of all phagosomes in a field of view by the number of fluorescent cells counted (i.e. phagocytic cells). (F) Average number of phagosomes per phagocytic cell. (G) The persistence of the cell during the length of its track. X-axis labels denote whether the cells were in media or confined and if there were beads present in the well. The last two columns were the same fields of view as counted in the middle two columns, but tracking was conducted only after a cell phagocytized at least one pHrodo-IgG-bead. This also applies for H and I. (H) Velocity of the cells migrating (µm/min). (I) The maximum accumulated distance (µm) that the cell traveled over the track. (J) Projections of single cell tracks from an example N, demonstrating all tracks, or post-phagocytic tracks only, in media and confined conditions with beads present. For all graphs, black points demonstrate experiment means, colored points demonstrate individual cell values for each run. N=4 experiments for each graph; n=15 fields of view for each condition per experiment for phagocytosis (C-F) or n=50 cells for each condition per experiment for migration (G-I). Statistical analysis was performed utilizing the total sum n values per condition. For (C-F), statistical analysis was assessed using the Mann–Whitney tests: ns=not significant, *P<0.05. For (G-I), statistical analysis was assessed using via Kruskal–Wallis test with Dunn multiple comparisons: ns=not significant, *P<0.05, ****P<0.0001. Error Bars represent s.e.m.

treated cells under confinement can be accepted with reasonable confidence to be an effect of the inhibitor.

Arp2/3 complex disruption with the small molecule inhibitor CK-666 decreased the percent of phagocytic cells in both media and confined conditions compared to vehicle (Fig. 3A, quantified in Fig. 3B). Arp2/3 disruption decreased the phagosome area under confinement (Fig. 3C; Fig. S3L) and phagosome number in confined and unconfined settings (Fig. 3C; Fig. S3M). Arp2/3 complex disruption also impaired phagocytic priming under confinement, as evidenced by decreased migratory persistence in post-phagocytic cells (Fig. 3D; Movie 3). Conversely, Arp2/3 complex disruption only decreased cell velocity and distance traveled in the media condition, but not migratory persistence after phagocytosis (Fig. S3N,O). We interpret these results to mean that the Arp2/3 complex is crucial for phagocytic uptake, trafficking of targets to the phagolysosome, and phagocytic priming in confined settings. While phagocytic uptake was affected in the media condition, Arp2/3 loss of function in this context had less impact than it did under confinement. These results suggest that there may be varying degrees of compensation for loss of Arp2/3 complex function that are context dependent.

## The actin cytoskeleton and non-muscle myosin II facilitate phagocytic priming and uptake

Previous studies examining myosin II in confined environments have demonstrated that it is needed for cell velocity but not for directional responses to either chemotactic (Barbier et al., 2019) or haptotactic cues (Stinson et al., 2025). We hypothesized in light of this that myosin II-dependent contractility would be involved in phagocytic uptake, but not phagocytic priming under confinement. Inhibition of myosin II with blebbistatin decreased phagocytic cell number in both confined and unconfined settings (Fig. 4A;

quantified in Fig. 4B). However, confinement partially rescued phagocytosis in blebbistatin-treated cells, as judged by increased uptake in these cells compared to blebbistatin-treated cells in media. This points to myosin-independent cytoskeletal elements that are activated by confinement and compensate for loss of actomyosin contractility. Bead trafficking and phagosome formation in cells that take up beads were unaffected by blebbistatin treatment, as demonstrated by all phagosome measurements (Fig. 4C; Fig. S4A). We interpret this to mean that myosin II impairment specifically affects phagocytic cup formation rather than intracellular trafficking to the phagolysosome.

Contrary to our expectations, myosin II disruption decreased phagocytic priming under confinement, as evidenced by decreased post-phagocytic persistence of confined cells treated with blebbistatin (Fig. 4D; Movie 4), while also decreasing velocity and distance traveled in this setting (Fig. S4B). Conversely, unconfined cells treated with blebbistatin showed no defect in post-phagocytosis persistence, again suggesting that myosin II's contribution to phagocytic responses is context dependent (Fig. 4D). To investigate the impact of beads on the blebbistatin phenotypes, we repeated these experiments without beads. Blebbistatin-treated cells moved similarly to their DMSO treated counterparts in terms of velocity, persistence, and distance traveled when comparing within the confined and unconfined groups when beads were absent (Fig. S4C). These findings differ from our findings when pHrodo-IgG beads were present. Phagocytic uptake could cause cells to patrol their environments more deliberately, especially under confinement, in a much more myosin II-centric fashion.

Inhibition of either the Arp2/3 complex or myosin II did not fully impair phagocytosis under confinement, pointing towards another component playing a compensatory role during confined phagocytosis. We next treated cells with cytochalasin D to inhibit new actin filament formation, examining the necessity of filamentous actin (F-actin) in phagocytic uptake and priming under confinement. Cytochalasin D treatment resulted in a significant decrease in confined and unconfined phagocytosis (Fig. 4E, quantified in Fig. 4F). Remarkably, confinement partially rescued phagocytosis in the presence of cytochalasin D. Phagosome formation, area, and number per phagocytic cell were all similarly affected by cytochalasin D, again with partial rescue by confinement (Fig. 4G; Fig. S4D). One interpretation of these findings is that the agarose gel, or its temperature at the time of cytochalasin D addition, partially inactivated the drug. However, soaking cytochalasin D-infused agarose gels with an equal volume of media and then treating unconfined cells with this 'conditioned' media demonstrated an inhibition effect similar to media directly treated with cytochalasin D (Fig. S4E). Given the fundamental importance of actin in phagocytic cup formation and endocytic trafficking, these results are not unexpected. However, these data suggest that in confined conditions a less efficient actin-independent phagocytosis pathway may exist.

It was less straightforward to interrogate phagocytic priming during cytochalasin D treatment. Cytochalasin D-treated cells were dramatically less migratory, and traveled far shorter distances than DMSO-treated counterparts in unconfined and confined settings (Fig. S4F, Movie 5). As already mentioned, cytochalasin D-treated cells were far less phagocytic than DMSO-treated counterparts (Fig. 4F). Because fewer post-phagocytic cells were therefore available for analysis, it is difficult to interpret the lack of significance between DMSO- and cytochalasin D-treated post-phagocytic persistence under confinement (Fig. 4H). However,

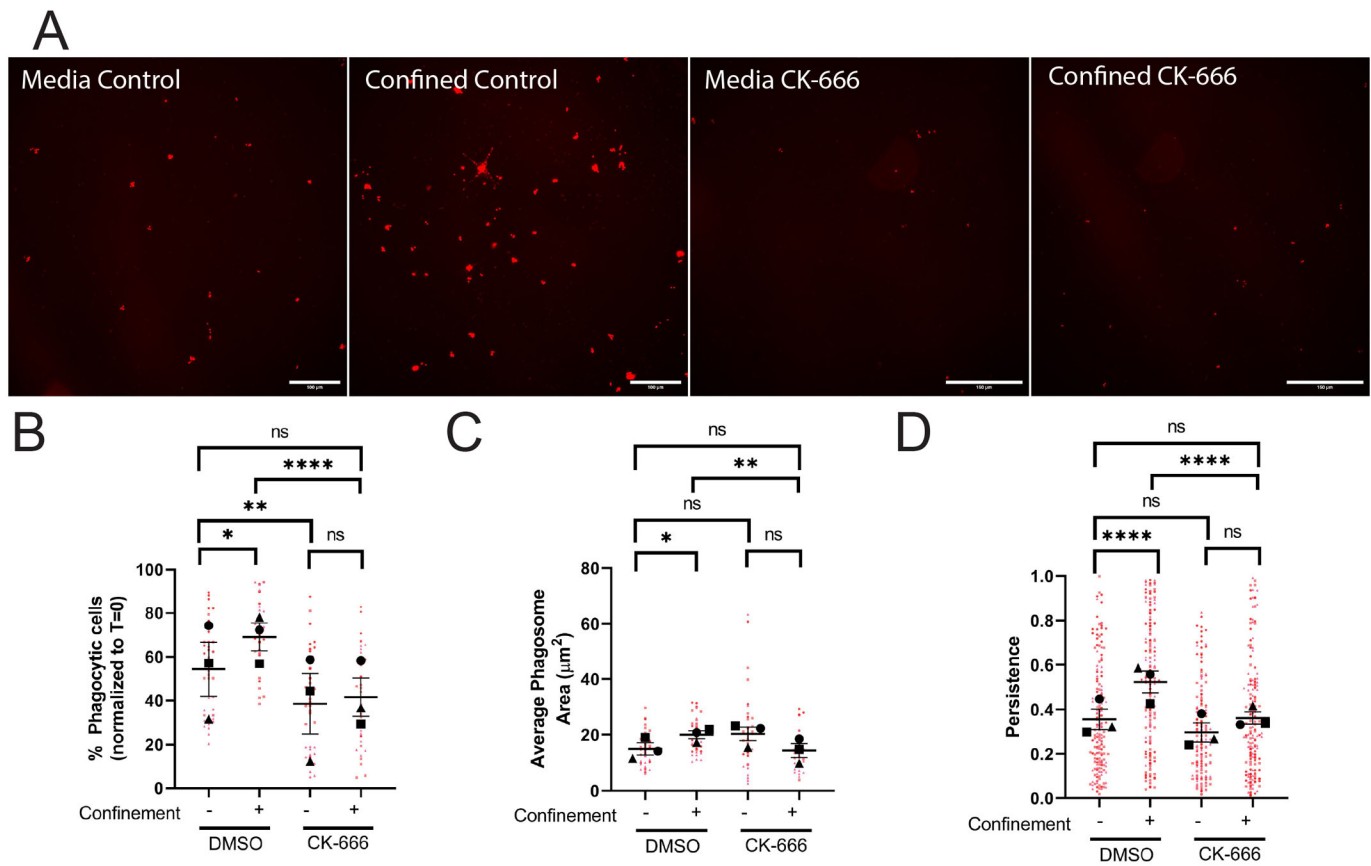

**Fig. 3. The Arp2/3 complex is required for phagocytosis and phagocytic priming in response to confinement.** (A-D) These cells were either treated with DMSO or 125 µM CK-666. (A) Example images of pHrodo-red label demonstrating internalized pHrodo-IgG-beads in each condition: Media control (vehicle – DMSO), Confined control (1% Agarose) plus Vehicle, Media control plus CK-666, Confined (1% Agarose) plus CK-666. Scale bars: 100 µm. (B) The percentage of fluorescent cells in a field of view, normalized to T=0. (C) Average phagosome size (µm²). (D) The persistence of post-phagocytic cells during the length of their migration track. For all graphs, black points demonstrate experiment means, colored points demonstrate individual cell values for each run. *N*=3 experiments for each graph; *n*=15 fields of view for each condition per experiment for phagocytosis (B,C) or *n*=50 cells for each condition per experiment for migration (D). Statistical analysis was performed utilizing the total sum *n* values per condition across all N. Statistical analysis was assessed using Kruskal–Wallis with Dunn multiple comparisons: ns=not significant, **$P<0.01$, ****$P<0.0001$. Error bars represent s.e.m..

since myosin II and Arp2/3 complex inhibition affected phagocytic priming, it is reasonable to suspect that phagocytic priming would be impacted if F-actin assembly is dramatically compromised. Given that Arp2/3 complex (Wu et al., 2012) and myosin II (Chabaud et al., 2015) are both effectors of cell adhesion, we next interrogated the role of ECM-integrin engagement as an additional factor regulating phagocytic uptake and priming under confinement.

### Phagocytic priming, but not uptake, is dependent on ECM interaction

We started by using cilengitide to inhibit αv-containing integrins. Cilengitide would be expected to impair BV2 interaction with the fibronectin substrate on the glass surface in confined and unconfined settings, as these cells express αvβ3 and αvβ5 integrins (Welser-Alves et al., 2011). Cilengitide treatment did not affect phagocytosis in either condition, compared to vehicle control (Fig. 5A,B). Overall, phagosome measures were similar between DMSO- and cilengitide-treated cells under confinement (Fig. 5C; Fig. S4G). Thus, inhibition of a subset of αv-containing integrins did not impact uptake or intracellular trafficking of pHrodo-IgG-beads.

However, cilengitide dramatically impacted phagocytic priming. Cilengitide-treated post-phagocytic cells were less persistent than

control cells in both settings (Fig. 5D; Movie 6). Our data also support the idea that cilengitide-treated cells may be less adhesive, as treated cells migrate faster and farther than control in both settings we assessed (Fig. S4H). These data suggest that phagocytic uptake and priming are not governed by identical pathways, as cilengitide-treated cells specifically have a defect related to post-phagocytosis migration rather than overall phagocytic ability. Since some integrin function is likely retained even in the presence of cilengitide, we were curious to see whether we could replicate these results by removing the response to ECM completely. To do this, we coated our dishes with 100 µg/ml PEG-Poly-L-Lysine (PEG-PLL) to limit integrin-mediated adhesion and compared cell responses to our standard coating of 10 µg/ml fibronectin. PEG-PLL elicits no change in phagocytosis compared to FN in either context (Fig. 5E,F; Fig. S4I). However, phagocytic priming was dramatically impacted by PEG-PLL. In both contexts, post-phagocytic persistence was significantly impaired when cells were plated on PEG-PLL compared to FN (Fig. 5H; Movie 7). Velocity and distance traveled increased on PEG-PLL under confinement, but not in media (Fig. S4J). All of these results were consistent with the results of cilengitide treatment, and demonstrated that ECM engagement was a key factor in phagocytic priming, especially under confinement, but not in phagocytic uptake.

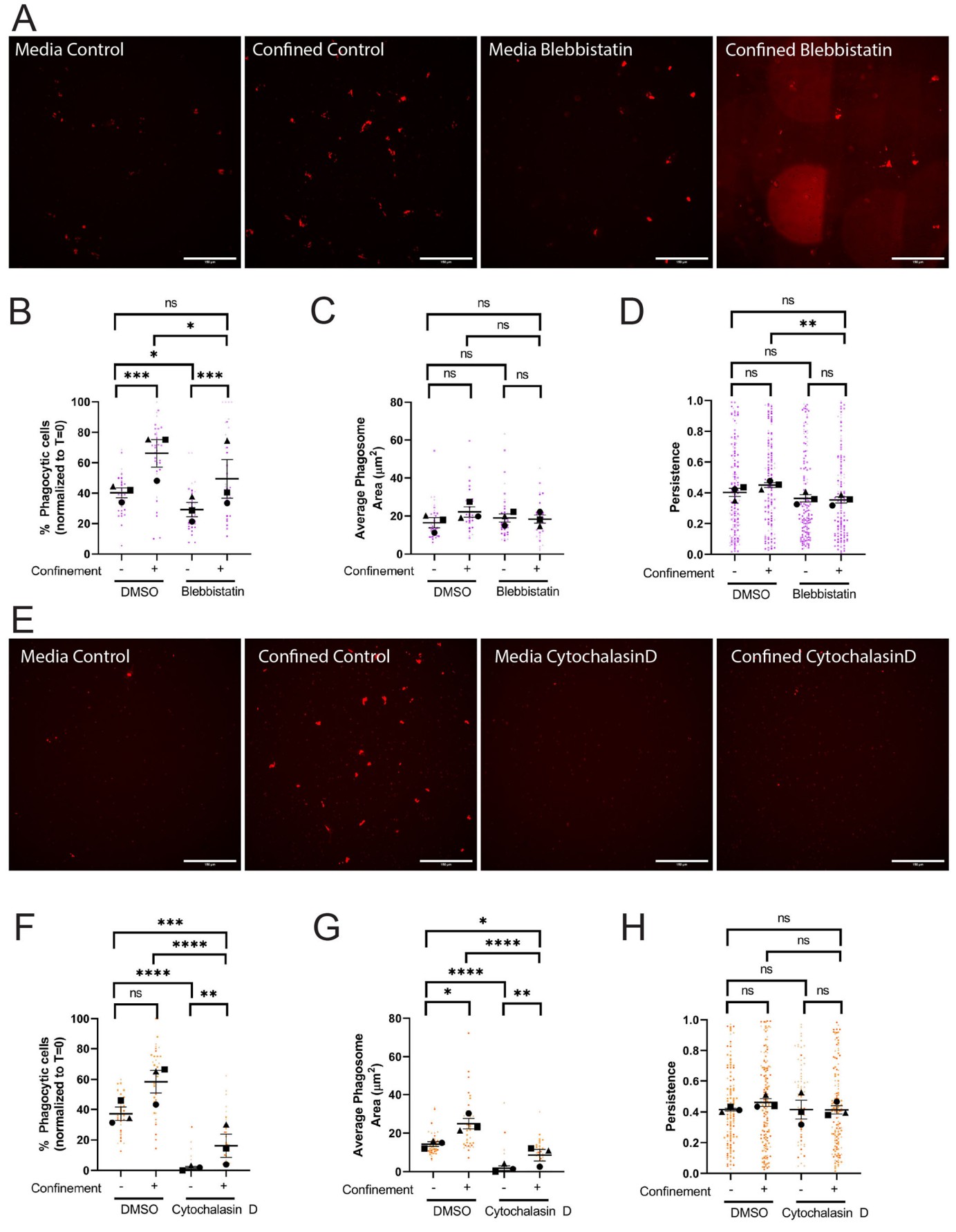

**Fig. 4.** See next page for legend.

**Fig. 4. Confinement enhances phagocytic uptake and priming via actomysosin.** (A-D) These cells were either treated with DMSO or 30 µM Blebbistatin. Blebbistatin caused strong auto-fluorescence when polymerized into agarose gels. These fluorescent fragments are visually distinct from beads in phase contrast. Composite images with both phase contrast and DsRed channels present were used to count phagocytic cells and manually remove blebbistatin autofluorescence from phagosome calculations. (A) Example images of pHrodo-red label indicating internalized pHrodo-IgG-beads in each condition: Media control (vehicle – DMSO), Confined control (1% Agarose) plus Vehicle, Media control plus Blebbistatin, Confined (1% Agarose) plus Blebbistatin. Scale bar: 100 µm. (B) The percentage of fluorescent cells in a field of view, normalized to T=0. (C) Average phagosome size ($\mu m^2$). (D) The persistence of post-phagocytic cells during the length of their tracks. (E-H) These cells were either treated with DMSO or 1 µM Cytochalasin D. (E) Example images of pHrodo-red label indicating internalized pHrodo-IgG-beads in each condition: Media control (vehicle – DMSO), Confined control (1% Agarose) plus Vehicle, Media control plus Cytochalasin D, Confined (1% Agarose) plus Cytochalasin D. Scale bars: 100 µm. (F) The percentage of fluorescent cells in a field of view, normalized to T=0. (G) Average phagosome size ($\mu m^2$). (H) The persistence of post-phagocytic cells during the lengths of their tracks. For all graphs, black points demonstrate experiment means, colored points demonstrate individual cell values for each run. $N$=3 experiments for each graph; $n$=15 fields of view for each condition per experiment for phagocytosis (B,C,F,G) or $n$=50 cells for each condition per experiment for migration (D,H). Statistical analysis was performed using the total sum $n$ values per condition across all N. Statistical analysis was assessed using the Kruskal–Wallis with Dunn multiple comparisons: ns, not significant; *$P$<0.05, **$P$<0.01, ***$P$<0.001, ****$P$<0.0001. Error Bars represent s.e.m..

## DISCUSSION

In the present study, we report that confinement increases phagocytosis, enabling more cells to take up IgG-opsonized beads compared to their unconfined counterparts. In addition, we discovered that phagocytosis had a 'priming' effect on cell motility, resulting in increased cell persistence after cells underwent an initial phagocytic event. Phagocytic priming was especially pronounced under confinement. Confined phagocytosis resembled unconfined phagocytosis, except that confinement is capable of partially rescuing phenotypes seen in unconfined settings. This led us to propose a two-part process that allows phagocytes to respond to phagocytic cues under confinement. First, confinement stimulates Arp2/3 complex and myosin II to take up IgG-beads more efficiently (Fig. 6). Then, particle internalization primes phagocytes to migrate persistently, which requires Arp2/3 complex, myosin II, and ECM engagement (Fig. 6). Our findings suggest that coherent interplay between mechanical sensing, ECM sensing, and phagocytic uptake determines cellular sensing and response to phagocytic targets. Some of these factors influence specific elements of this process, and the interplay between them and the cytoskeleton is worth exploring further.

Post-phagocytic cells under confinement consistently demonstrated persistent migration, which we term phagocytic priming. Disrupted phagocytic priming sometimes correlated with altered phagosome measurements (as with treatment with CK-666 or cytochalasin D), but other times phagosome measurements were unaffected by the lack of priming (as with treatment with blebbistatin, cilengitide, or cells plated on PEG-PLL). A study by Procès et al. found that BV2 cells activated via mechanical stretch became more directionally persistent in one-dimensional assays in PDMS microchannel devices (Procès et al., 2024), suggesting that mechanotransduction of external forces by the actomyosin cytoskeleton may be the underlying driver of phagocytic priming. This is in line with the results of this study: key drivers of phagocytic priming identified here, myosin II and the Arp2/3 complex, are responsive to mechanical force (Barbier et al., 2019;

Bieling et al., 2016; Liu et al., 2015; Lomakin et al., 2020; Mueller et al., 2017; Papalazarou and Machesky, 2021). In addition, stretch-induced persistent migration of BV2 cells activated Arp2/3 complex in lamellipodial protrusions via Rac GTPases (Procès et al., 2024). Procès et al. also noted increased persistence of BV2 cells after treatment with lipopolysaccharides (LPS) compared to mechanically activated counterparts, with both activation methods being more persistent than control cells (Procès et al., 2024). Previous studies have also demonstrated increased migratory persistence under confinement independent of an external gradient when examining cell migration in 3D environments using cancer cells (Irimia and Toner, 2009; Rolli et al., 2010) or fibroblasts as model systems (Petrie et al., 2009). These findings align well with our data, suggesting that persistent migration is inducible by physical force, and that pathogen sensing similarly cooperates with mechanical confinement to alter motility.

The results of this study also reinforce the idea that Arp2/3's influence is context dependent. Here, we show that Arp2/3 loss impacts IgG phagocytosis in confined and unconfined settings. However, post-phagocytosis persistence is only impaired under confinement, while migration velocity is only impaired in unconfined settings. This is in line with previous studies demonstrating Arp2/3's context dependence. Directional motility is impaired in a variety of Arp2/3-deficient cell types in haptotactic (Nicolai et al., 2020; Rotty et al., 2017; Wu et al., 2012), but not chemotactic settings (Davidson et al., 2018; Rotty et al., 2017; Vargas et al., 2016; Wu et al., 2012). The involvement of Arp2/3 in phagocytosis also depends on the opsonizing cue. For example, macrophages require Arp2/3 complex for iC3b-mediated phagocytosis, but Arp2/3 is not as important for IgG phagocytosis (Rotty et al., 2017). However, this could be cell-type dependent, as we demonstrate a greater need for Arp2/3 in BV2 mediated phagocytosis. Macrophages do not require Arp2/3 to maintain directional migration as they transition into under agarose confinement (Stinson et al., 2025), but microglia do need it to maintain persistent migration after phagocytosis under confinement. This could be due to different migratory requirements in these two cell types, but it could also be related to the unique environmental context of these two populations as well. It will be interesting to determine whether other phagocytes are also primed by phagocytosis to migrate in a persistent fashion, and whether evidence of this response can be detected *in vivo*.

IgG opsonizes foreign bodies in the context of infection and can initiate tissue inflammatory immune pathways by binding to Fc gamma receptors (FcγRs) expressed on the plasma membrane (Castro-Dopico and Clatworthy, 2019). Phagocytosis of IgG-bound particles aids in clearance of pathogens and foreign debris. One example of this *in vivo* is immature dendritic cells that surveil tissue until they encounter an antigen (possibly bound by IgG), at which point they mature and follow a gradient of CCL21 to the lymph nodes in order to incite a T cell response to the pathogen (Hampton and Chtanova, 2019; Liu et al., 2021). Phagocytic priming could enable cells to move more efficiently to the lymphatics, as well as better managing pathogen control in peripheral tissues. The impact of IgG on microglial function in the central nervous system is complicated. Un-complexed IgG is proposed to have neuroprotective effects, via stimulation of microglial endocytosis (Hulse et al., 2008). However, IgG-ligand complexes (as modeled by our IgG beads) have been proposed to exacerbate detrimental microglial responses in multiple sclerosis (van der Poel et al., 2020), while IgG-amyloid beta complexes are commonly present within amyloid plaques (Kellner et al., 2009). It is possible that Arp2/3-dependent, ECM-responsive haptotaxis helps explain the response to these IgG complexes, as it has been implicated previously in haptosensing (Rotty et al., 2017).

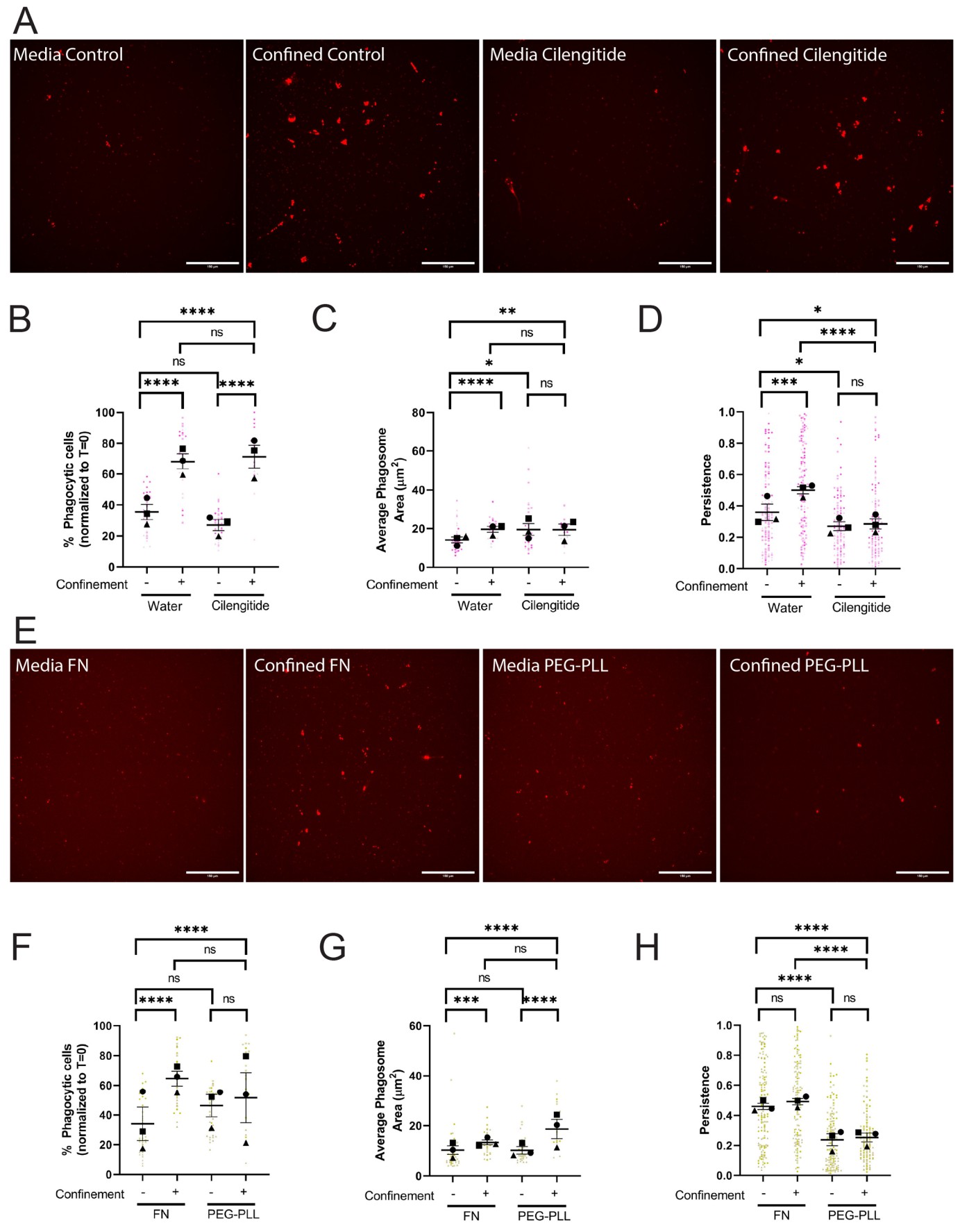

**Fig. 5.** See next page for legend.

**Fig. 5. Cell adhesion is required for confinement-induced phagocytic priming, but not uptake.** (A-D) These cells were either treated with water or 60 μM Cilengitide. (A) Example images of pHrodo-red label indicating internalized pHrodo-IgG-beads in each condition: Media control (vehicle – water), Confined control (1% Agarose) plus Vehicle, Media control plus Cilengitide, Confined (1% Agarose) plus Cilengitide. Scale bars: 100 μm. (B) The percentage of fluorescent cells in a field of view, normalized to T=0. (C) Average phagosome size ($μm^2$). (D) The persistence of post-phagocytic cells during the length of its track. (E-H) These cells were either coated on 10 μg/ml FN or 100 μg/ml PEG-PLL. (E) Example images of pHrodo-red label indicating internalized pHrodo-IgG-beads in each condition: media control (vehicle – FN coating), confined control (1% Agarose) plus vehicle, media control plus PEG-PLL coating, Confined (1% Agarose) plus PEG-PLL coating. Scale bars: 100 μm. (F) The percentage of fluorescent cells in a field of view, normalized to T=0. (G) Average phagosome size ($μm^2$). (H) The persistence of post-phagocytic cells during the lengths of their tracks. For all graphs, black points demonstrate experiment means, colored points demonstrate individual cell values for each run. $N$=3 experiments for each graph; $n$=15 fields of view for each condition per experiment for phagocytosis (B,C,F,G) or $n$=50 cells for each condition per experiment for migration (D,H). Statistical analysis was performed using the total sum $n$ values per condition across all N. Statistical analysis was assessed using the Kruskal–Wallis with Dunn multiple comparisons test correction: ns=not significant, *$P$<0.05, **$P$<0.01, ****$P$<0.0001. Error bars represent s.e.m.

Phagocytic priming may help explain how microglia find these IgG complexes, and why they remain associated with lesions and plaques *in situ*. Future studies will also reveal whether phagocytic priming is a universal response to phagocytosis or unique to IgG. It is also imperative to examine whether primed cells are more responsive to chemotactic or haptotactic cues.

Paterson and Lämmermann recently discovered that macrophages manage phagocytosis via an integrin-dependent process termed haptokinesis (Paterson and Lämmermann, 2022). When following cellular behavior in a 3D Matrigel context, removal of β1 integrin or talin significantly impaired motility, which disrupted macrophages' ability to contact phosphatidylserine beads or apoptotic cells to internalize them. Of note, they also demonstrated that β1 integrin or talin disruption did not impair phagocytosis when cells were in suspension. Consistent with this, we demonstrate no defect in phagocytic uptake upon cilengitide treatment or removal of

fibronectin. We also show that there is an impact on motility in the context of phagocytic priming. These results are largely consistent with Paterson and Lämmermann's findings, with the caveat that cells in our studies were likely have more freedom to interact with beads than in 3D Matrigel. In addition, our motility measurements were focused on post-phagocytic behavior, which necessarily selects for cells that have successfully taken up beads regardless of treatment.

The present work is consistent with the idea that integrating microenvironmental factors in a controllable fashion is key to understanding complex cellular behaviors. It reveals that an intersection of mechanical cues, adhesion cues, and phagocytic uptake together stimulate a two-part response to IgG beads, driven by differential cytoskeleton regulation. This integration of multiple microenvironmental factors is important in aligning a cell's behavior in an *in vitro* setting with its *in vivo* function. Over time it will be possible to model microglia *in vitro* in a way that more completely captures their *in vivo* morphology and behavior, which is important to further understand their role in neurological diseases and to reveal future therapeutic targets (Asai et al., 2015; Frakes et al., 2014; Hulse et al., 2008; Kellner et al., 2009; Lee et al., 2010; Liao et al., 2012; Maezawa and Jin, 2010; van der Poel et al., 2020).

## MATERIALS AND METHODS
### Cells
The murine microglia-like BV2 cell line was used for all experiments, and were received from Dr Gabriela Dveksler (Uniformed Services University). Cells were grown in complete cell culture media, containing Dulbecco's modified Eagle's medium (DMEM) (Gibco, 31053028), 5% fetal bovine serum (FBS) (Gibco, 10437028), 1% glutaMAX (Gibco, 35050061), and 1× antibacterial-antimycotic (Gibco, 15240062) at 37°C and 5% $CO_2$. 70-80% confluent dishes were treated with 0.05% Trypsin–EDTA solution (Caisson Labs TRL02-100ML) at 37°C for 10 min. The solution was then aspirated and replaced with 2 ml of complete cell culture media that was gently sprayed over the bottom of the dish to dislodge cells and collect for counting and passage into new dishes.

### Four-well chamber preparation
Four-well chambers (Cellvis, C4-1.5H-N) were coated in 10 μg/ml fibronectin (Gibco, 33016015) for 1 h at 37°C. Chambers were washed

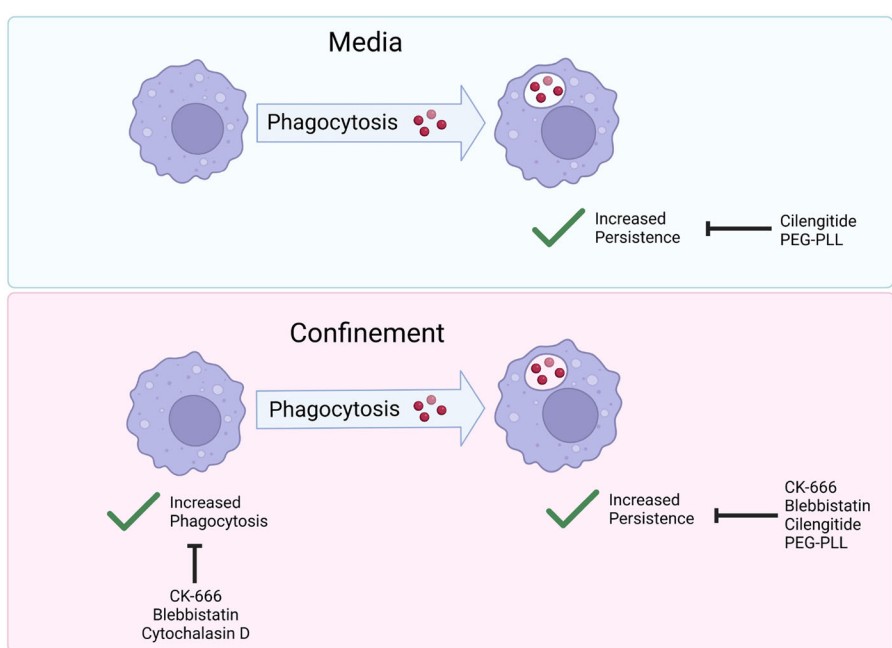

**Fig. 6. The actin cytoskeleton and cell-ECM adhesion respond to confinement and phagocytic targets to tune phagocytic uptake and priming.** Graphical abstract summarizing paper findings. Cells in media experience increased persistent migratory patterns. These patterns are disrupted when integrin binding is blocked (by Cilengitide), or fibronectin is replaced by PEG-poly-L-lysine (PEG-PLL). Cells under confinement experience increased phagocytosis and increased phagocytic priming. Increased persistence is mediated by myosin II (inhibited by blebbistatin), Arp2/3 complex (inhibited by CK-666) and integrin-ECM engagement (inhibited by cilengitide or PEG-PLL). Increased phagocytosis is mediated by Arp2/3 complex and myosin II (inhibited by CK-666 and blebbistatin, respectively). Created in BioRender by Paulson, S. (2025) https://BioRender.com/rigafy4.

three times with 1× Cell Culture Phosphate Buffered Saline (Corning, MT21040CV) (referred to from here on as PBS). Two wells were each filled with 1 ml PBS (for later replacement by complete cell culture media) and two wells were each filled with 1 ml 1% agarose. Agarose was prepared as follows: 2.5 ml of 2× Hank's balanced salt solution (Sigma, H1387-10X1L) and 5 ml of complete cell culture media (as defined above) were combined and placed in a 68°C bead bath for 1 h. 0.12 g of low gelling temperature agarose (Sigma, A9045-5G) was added to 2.5 ml of sterile water (Corning, 25-055-CV) and heated in a microwave until fully dissolved. The two solutions were combined to create a warm 1% agarose mixture. 1 ml of the mixture was poured into each well and allowed to solidify for 90 min at room temperature. Chambers were wrapped in parafilm and stored at 4°C for up to 1 month.

### Phagocytosis bead preparation

100 µg of Normal Mouse IgG (Sigma-Aldrich, 12-371) was pHrodo labeled using the pHrodo™ iFL Red Microscale Protein Labeling Kit (ThermoFisher Scientific, P36014) according to manufacturer instructions. 60 µl of pHrodo-IgG was opsonized to 60 µl of 2-micron polybead carboxylate microspheres (Polysciences, Inc, 18327-10) (hereafter referred to as beads) in 3 ml of 1× PBS at 37°C for 1 h. Beads received three washes with 1× PBS after completing opsonization and were resuspended in a final volume of 500 µl, stored at 4°C for up to 4 months. Before usage in an experiment, beads were vortexed vigorously.

### Cell preparation

Agarose wells had two punches placed in each for cell and bead insertion using a 0.75 mm biopsy punch tool (LabTech, 52-004908). Unconfined media wells had PBS replaced with complete cell culture media in preparation for cell seeding. Confluent cell culture dishes were treated according to the standard passaging protocol (above). For cell migration experiments, 13,000 cells were added to each media well. For phagocytosis, 18,000 cells were added to each media well. Lower numbers were used for migration experiments to minimize cell–cell interactions and allow more room for uninterrupted random migration. For both types of experiments in the confined condition, 15,000 cells were inserted under the agarose in each punched location using a gel-loading pipette tip. If the volume of media containing 15,000 cells exceeded 15 µl, the cells were spun down at 1000× $g$ and resuspended in 10 µl of complete cell culture media. Chambered dishes were placed at 37°C for 3 h to allow for cells to sit down and spread before being moved to the microscope. For the inhibitor trials, each cell culture media well received either vehicle or inhibitor to create the working concentration used for cell treatment (more information below).

### Phagocytosis microscope assay

Upon completing the 3-h incubation, IgG beads were added to wells. Media wells received 5 µl of beads. For confined wells, a 1:4 dilution of the beads was made with PBS and inserted into each punch in the agarose to match bead density with the media condition. Chambers were then placed in a Tokai Hit INU incubation system controller on the Olympus IX83 and maintained at 37°C, 90% humidity, and 5% $CO_2$ for the duration of the live cell imaging. A 20× air objective was employed during overnight time lapse imaging. Fifteen fields of view per well were selected. Images were taken at each field of view every 10 min for the span of 8 h. Both the relief contrast and the DsRed channels were used. The DsRed channel was set to 500 ms exposure to detect pHrodo signal upon bead internalization by cells.

### Analyzing bead density

Each phagocytosis file was opened in Fiji ImageJ. A bandpass filter was then run on the relief contrast channel. Structures were filtered to fit between 5 and 15 pixels and then threshold filtered until only the beads were highlighted in red. The analyze particles function was used to count bead density. Area threshold was set to 10-30 micron$^2$, circularity threshold was set to 0.8-1.0, and outlines were shown. When summarized, average size was maintained between 15 and 17 micron$^2$ across the time lapse, with threshold being adjusted and analysis rerun if the averages were too small or too large. Counts of beads per time point were averaged to create the bead

density designator for each file. Averaged bead numbers falling between 50 and 500 were designated as low bead density; between 500 and 950 as medium bead density; and between 950 and 1350 as high bead density. To control for variable bead densities between conditions, the number of beads present in media wells was first quantified, with corresponding bead densities in confined wells used for comparison.

### Analyzing phagocytic rate

Each phagocytosis file was opened in FIJI ImageJ. Channel colors were corrected and brightness/contrast adjusted to create the same brightness and contrast parameters for each file. Using the cell counter plugin, the total number of fluorescent and non-fluorescent cells was counted at each measured time point throughout the phagocytosis videos. Comparisons between treatment groups at individual time points as well as comparing changes over time within each group were conducted with GraphPad Prism. Graphs were normalized to T=0. Fluorescent cells at the beginning of the videos were removed from the count of both fluorescent cells and total cells per field of view. Data comparisons were broken up by bead density, only comparing media and confined fields in the low bead density category to each other. Blebbistatin auto-fluoresces under fluorescent imaging. Visual inspection of phagocytic cells ensured that counted fluorescent internalized pHrodo-IgG and not blebbistatin.

### Analyzing phagosome size

Each phagocytosis file was opened in CellSens Dimension software under the Count and Measure plugin. Under detection options, minimum object size was set to 5 pixels. Under the Thresholding tab, the adaptive threshold was selected. Selecting the DsRed channel, the maximum was set to 65,000 (the maximum) and the minimum was changed for each image to label only internalized beads, although never going below 1000. Selecting the relief contrast channel, the maximum was set to 2 and the minimum to 1 so that only fluorescent labels were detected. Outputs of total number of phagosomes and their average size per field of view were then verified at each timepoint before the data was exported to an excel file. This data was presented three ways. The average size of phagosome was reported by dividing the average size per field of view at 2 h by the total number of phagosomes detected at 2 h. Average size of phagosome per phagocytic cells was reported by dividing the average size per field of view by the number of phagocytic cells at that timepoint. The average number of phagosomes per phagocytic cell was reported by dividing the total number of phagosomes at 2 h by the number of phagocytic cells at that timepoint. Average size of phagosome and average size of phagosome per phagocytic cell differ as the former examines whether beads are trafficked into the same or different lysosomes and the latter examines if more beads overall are being trafficked in each treatment condition.

### Analyzing motility

Each motility file was opened in FIJI ImageJ. The Manual Tracking plugin was used to track 10 cells per field of view, using the base of each cell's forward-most protrusion, and data was combined across all fields of view within treatment groups. Cell tracking was stopped if a cell divided or ran into another cell, resulting in a changed direction. To obtain post-phagocytosis motility measurements, tracks were obtained only after a cell phagocytosed at least one pHrodo-IgG bead. In the case of cytochalasin D trials, there were far fewer cells taking up beads in the presence of drug. Thus, not as many cells in these trials fit the inclusion criteria, thereby accounting for the smaller number of cells analyzed for motility in this condition. The resulting measurements were uploaded into the Ibidi Chemotaxis and Migration Tool ImageJ plugin to measure velocity, accumulated distance, and persistence. Persistence (also termed D/t) is a measure of the straight-line distance between a cell's starting and ending position on a track (D) divided by its overall track length (t). Thus, a value very close to zero (high t, low D) is understood to be meandering much more than a cell moving along the shortest distance between its start and end (d=T), which has a value of 1. The closer the value to 1, the more persistent the migration is. These data tables were exported to GraphPad Prism to graph differences in velocity, distance, and persistence between treatment groups.

## Different agarose concentrations

Four-well chamber dishes were coated with fibronectin, as described above. These chambered dishes were made to contain one well of PBS, one well of 1% agarose, one well of 2.5% agarose, and one well of 5% agarose. 1% agarose was prepared as stated above. For both the 2.5% and the 5% agarose concentrations, only the amount of agarose added to the sterile water differed in the protocol. 2.5% agarose was prepared using 0.3 g of agarose in 2.5 ml of sterile water and 5% agarose was prepared using 0.6 g of agarose in 2.5 ml of sterile water. Chambers were wrapped in parafilm and stored at 4°C for up to 1 month.

## Microscope setup for motility assay

Upon completing the 3-h incubation, chambers were placed in an environmental chamber on a Tokai Hit INU incubation system controller on the Olympus IX83 and maintained at 37°C, 90% humidity, and 5% $CO_2$ for the duration of the live cell imaging. A 20× air objective was employed during overnight time lapse imaging. Ten fields of view per well were selected. Images were taken at each field of view in the relief contrast channel every 10 min for the span of 16 h.

## Inhibitor-containing agarose

CK-666 (abcam, ab141231), Blebbistatin (ThermoFisher Scientific, NC0664123), and Cytochalasin D (ThermoFisher, PHZ1063) were resuspended in anhydrous DMSO (ThermoFisher Scientific, D12345) to create stock solutions. Cilengitide (Sigma Aldrich, SML1594-5MG) was resuspended in water (Corning, 25-055-CV) to create a stock solution. Agarose was prepared as previously described and then split into two aliquots of 5 ml warm agarose solution. Each aliquot received either vehicle (anhydrous DMSO or water) or the drug stock solution at a predetermined concentration (CK-666 125 µM, Blebbistatin 30 µM, Cytochalasin D 1 µM, Cilengitide 60 µM). Agarose chambers were prepared as previously described, wrapped in parafilm, and stored at 4°C for up to 1 month. For PEG-Poly-L-Lysine (PEG-PLL) (Creative PEG Works, PPL-2k20k3.5-100 mg) plates, two wells were coated with 100 µg/ml PEG-PLL for 1 h at room temperature, then 10 µg/ml fibronectin were added to the two non-coated wells and coated at 37°C for 1 h. Agarose was then prepared as previously described. For the Cytochalasin D gel soaking experiment, Cytochalasin D containing agarose was made as previously described and seeded into a six-well dish (ThermoFisher Scientific, 140675) at 2 ml per well. An equivalent amount of complete cell media was added to the well after the agarose had solidified, and the dish was placed in an incubator for 12 h. Media was harvested at 12 h from the top of the gel and used in a four-well chamber dish, then compared to wells containing vehicle (DMSO), 0.5 µM, or 1 µM Cytochalasin D. To rule out an effect of HBSS on cell phagocytosis, agarose media was made up containing DMEM, HBSS, and water equal to the composition of the agarose gel. Unconfined cells were treated with this HBSS containing media and compared to cells in the standard (HBSS-containing) under agarose condition.

## Inhibitor phagocytosis and motility runs

Cells were prepared as previously described. Media wells contained either vehicle or drug at the same concentration present in the agarose. Cells were seeded as previously described and plates run on Olympus IX83 as previously detailed.

## Quantification of unconfined and confined cell height

Four-chamber well dishes were created as previously described, with the addition of Cascade Blue-labeled 3000 M.W. dextran (ThermoFisher Scientific, D7132) at 10 µg/ml to the agarose prior to pouring. Cascade Blue dextran was also added to the media for unconfined cells (at 5 µg/ml) prior to imaging, and beads were added to the agarose well 2 h prior to imaging. A Nikon inverted spinning disk confocal microscope outfitted with an EMCCD camera (Yokogawa) and 60× Plan Apo 1.4 NA oil objective was used to image the BV2 cells. NIS-Elements software controlled the microscope during image acquisition. Z-sections were taken at 0.3-micron intervals for a total of 26 µm per image. Fiji ImageJ was used to count the number of Z-sections in each cell and to construct the side projections of cells. Cell heights were calculated by multiplying the number of steps required to encompass the cell by the step size (0.3 µm). Despite the cells

being unlabeled, due to our BV2 cells not tolerating tracker dye well, cell edges could be clearly defined and differentiated from the dextran in both the media and agarose.

## Statistical analysis

The Kruskal–Wallis with Dunn multiple comparisons test was used to assess significance in experiments where a normal distribution of the dataset could not be assumed. When only two experimental conditions were tested, we used Mann–Whitney tests when we could not assume normality. Unpaired $t$-tests and ANOVAs were used when normality tests indicated a normal distribution of the data. All statistics were calculated using GraphPad Prism, and significance was assumed if $P \leq 0.05$. More information on each statistical test can be found in the relevant figure legend panel.

## Acknowledgements

We thank the members of the Rotty Lab for helpful discussions during research development, to Dr Gabriela Dveksler for access to BV2 cells, and to Dr Prasanna Satpute-Krishnan for assistance with live cell confocal imaging. This project is sponsored by the Uniformed Services University of the Health Sciences (USU); however, the information or content and conclusions do not necessarily represent the official position or policy of, nor should any official endorsement be inferred on the part of, USU, the Department of Defense, or the US Government.

## Competing interests

The authors declare no competing or financial interests.

## Author contributions

Conceptualization: S.G.P., J.R.; Data curation: S.G.P., S.L., J.R.; Formal analysis: S.G.P., S.L., J.R.; Funding acquisition: J.R.; Investigation: S.G.P., J.R.; Methodology: S.G.P., J.R.; Project administration: J.R.; Supervision: J.R.; Validation: S.G.P., J.R.; Writing – original draft: S.G.P., J.R.; Writing – review & editing: S.G.P., S.L., J.R.

## Funding

This work was supported by a Uniformed Services University graduate student research award (to S.G.P.), a Cosmos Club Foundation award (to S.G.P.), and by the National Institutes of Health (GM134104, to J.R.), Department of Defense (HU00012320103, to J.R.), and startup funds from the Uniformed Services University (to J.R.). The Uniformed Services University of the Health Sciences (USU), 4301 Jones Bridge Rd., A1040C, Bethesda, MD 20814-4799 is the awarding and administering office. Open Access funding provided by NIH and DoD. Deposited in PMC for immediate release.

## Data and resource availability

All primary data will be openly available upon request. All relevant data and details of resources can be found within the article and its supplementary information.

## Peer review history

The peer review history is available online at https://journals.biologists.com/bio/lookup/doi/10.1242/bio.062021.reviewer-comments.pdf

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
