## [Peer Review File · Biology Open]

Physical confinement and phagocytic uptake induce persistent cell migration

Summer G. Paulson, Sophia Liu and Jeremy Rotty
DOI: 10.1242/bio.062021

Editor: Catherine L. Jackson

Review timeline

Original submission:	14 April 2025
Editorial decision:	23 April 2025
First revision received:	31 July 2025
Editorial decision:	8 August 2025
Second revision received:	12 August 2025
Accepted:	13 August 2025

Original submission

First decision letter

MS ID#: bio.062021

MS Title: Physical confinement and phagocytic uptake induce persistent cell migration

Authors: Summer G. Paulson, Sophia Liu and Jeremy Rotty

I have now reached a decision on the above manuscript.

The reviewer reports are shown at the bottom of this email or can be accessed, together with a copy of this decision letter, by going to:

As you will see, the reviewers gave favourable reports, but raised some critical points that will require amendments to your manuscript. I hope that you will be able to carry these out, because we would like to be able to accept your paper.

At this stage, we also ask you to ensure your manuscript complies with our formatting guidelines “please see our manuscript preparation guidelines for details. Provided you are able to fully address the referees’ comments, we are positive about publication of your paper (we accept over 95% of revision submissions) and therefore hope you won’t mind any extra work involved in reformatting your manuscript at this point.

Please ensure that you clearly highlight all changes made in the revised manuscript. Please avoid using ‘Tracked changes’ in Word files as these are lost in PDF conversion.

I should be grateful if you would also provide a point-by-point response detailing how you have dealt with the points raised by the reviewers in the ‘Response to Reviewers’ box. Please attend to all of the reviewers’ comments. If you do not agree with any of their criticisms or suggestions please explain clearly why this is so.

Reviewer 1

Comments for the author

This manuscript from Paulson, Liu, and Rotty describes their efforts to define how physical confinement impacts phagocytosis and motility in microglial cells. The studies use the well-established immortalized mouse line, BV-2. They find that confinement, wherein the BV-2 cells are essentially sandwiched between the underlying substrate and an agarose blanket, promotes phagocytosis of fluorescently labeled IgG. They go on to show, using a collection of validated pharmacological tools, that the Arp2/3 complex and myosin II are required for confinement-mediated phagocytosis. Furthermore, they demonstrate that phagocytosis induces cell motility, which they term "phagocytic priming." Finally, they show that phagocytic priming (cell motility) requires integrin-based adhesion. Overall, this study provides insight into the mechanics of microglial phagocytosis and how phagocytosis promotes cell motility.

The experimental quality is good. The figures, including supplemental, are straightforward to interpret. The assays used are appropriate for addressing the questions asked (e.g., phagocyte formation) and include the proper controls.

Reproducibility is addressed. All experimental data values are plotted along with biological replicate means. At least three replicates are shown for all experiments. The appropriate parametric or non-parametric statistical tests were used based on the data distribution. Necessary details are provided in both the Methods and Figure Legends.

The presented studies are defined and complete. The data largely support the conclusions, though I have suggestions for improvement (mostly on directionality, see below).

Overall, the scholarship is good. The text is clear and well-written. The authors discuss their findings appropriately, citing existing work and incorporating this study into the larger body of work on phagocytosis and cell movement (though I do take issue with one discussion point - see below)

Major concerns

1. My primary concern is how the authors use persistence and directionality interchangeably. Persistence is not a measure of directionality. Directionality is a bias in one direction (i.e., the movement of a cell toward a chemotactic or haptotactic cue). In contrast, persistence is how long a cell moves in the same direction. The experimental analysis of cell motility used in this study can provide information on persistence, not directionality. Perhaps I'm missing something here, but the studies do not address haptotaxis or chemotaxis, which could provide insight into directionality. Additional studies are unnecessary, but the authors should be cautious when describing and interpreting the current data.
2. Along those lines, Figure 1B is misleading as it portrays cell movement in response to a gradient (haptotactic?), but this was not tested/measured.

Minor concerns

1. Quantifying the phagocytosis experiments required a subjective cutoff to account for variability in bead distribution between substrates. This is okay, but I would like more information on how this was done in the Methods.
2. While reading this, I questioned if physical confinement has been shown to regulate persistent movement in other cell types. It turns out it does, as the authors mention in the Discussion. This is an important point and should be raised earlier in the text.
3. I was surprised that blebbistatin treatment did not affect cell movement without beads. Were the authors surprised by this?
4. Could the confinement phenotypes observed with CytoD be caused by differences in uptake (thus altering the effective concentration)? Did the authors try different (higher) concentrations of CytoD? At lower concentrations, CytoD can limit barbed end growth (mimicking capping protein) rather than disrupt the entire actin cytoskeleton.

5. Cilengitide experiments. Perhaps I missed it, but did the authors confirm that alphaVbeta3 or alpha5beta5 integrins (the targets of Cilengitide) are the primary integrins expressed in BV-5 cells?
6. The discussion on the MTOC was unwarranted as no data was provided to support a change in MTOC position. I recommend removing this from the Discussion.
7. Figure 2 legend. I was confused by the description of how the average phagosome area and average number of phagosomes per cell were calculated. For example, "Average phagosome area per phagocytic cell. This was calculated by dividing the average size of all phagosomes in a field a (of - typo) view by the number of fluorescent cells counted (i.e. phagocytic cells)." Is this correct? Or did you divide the total phagosome area (as measured by fluorescence) in the field by the number of phagocytic cells? These details could also be moved to the Methods.
8. I was surprised that CK-666 did not disrupt persistence as it does in other cell types (like fibroblasts), though it did reduce cell speed (consistent with past data). Might microglia be less dependent on Arp2/3 for cell movement? Should discuss further in the Discussion.

Reviewer 2

Comments for the author

The impacts of physical confinement on cell migration have been studied extensively in recent decades, and it is now widely accepted that cells in confinement often migrate differently than cells on coverslips. Despite this, there has been little to no attention paid to how confinement can influence phagocytosis. This is surprising, given that many of the actin-dependent molecular mechanisms that drive migration also drive phagocytosis. This manuscript by Paulson et al expertly identifies this gap in our collective thinking about cells in physiological contexts. The authors find that confinement increases the rate of phagocytosis, and following phagocytosis cells move with increased directional persistence—a phenomena the authors name "phagocytic priming." Overall, this is an important study that could change the way we think about phagocytosis in physiological environments. While I am overall enthusiastic about this work (and I look forward to sharing this work with my colleagues for a journal club whenever it is public!), I have a few hesitations about some of the methods, conclusions about directional migration, and the statistical analyses.

Major Points:

1. Statistical questions: Were the statistical analyses performed on experiment-level averages (N=3-4) or on the entire dataset (dozens to hundreds of cells)? This information is not clear from the methods or legends. If the statistics were done on the entire dataset using the statistical tests as described, I have some concerns about pseudoreplicates artificially deflating the p values. It is also worth noting if the replicates with the same shape were completed on the same day. Finally, I think it would be useful to show the significance between confined and unconfined cells in DMSO for Fig. 3 B-D, Fig. 4 B-D and F-H, water in 5 B-D, and FN in 5 F-H.
2. Were the unconfined cells in complete media (as described in the Figure 1 legend, line 521), or PBS (as described in methods, line 393)? Either way, the agarose solution appears to be a bit different (HBSS and media as described in methods). Because these solutions are different, is it possible that the different osmolarities, salt concentrations, pHs, etc. could be driving the difference between unconfined and confined cells?
3. How confined are the cells? They do not look particularly flattened in movie 1, and the beads seem to be floating/flowing, suggesting the ceiling is higher than 2 microns. Is it possible to measure the height?
4. Is it possible the heat from the agarose inactivated some of the drugs? This could explain how an agarose pad rescued the Cytochalasin D phenotype, and would also explain the lamellipodia-like structures that seemed to be abundant in video S2 in the presence of CK-666. Some type of control to prove the drugs are still effective would be useful (some ideas: removing the agarose pad if

possible? Having cells crawl on top of the agarose pad instead? Making a gel with drugs at 2x, incubating in an equal volume of drug-free media, then testing that media on unconfined cells?).

5. Conclusions about directed migration. A central point of the manuscript (and the title) is that phagocytosis increases directional migration. For example, line 149 claims that "bead uptake seemed to induce more persistent migration tracks" with a reference to Movie 1. From watching movie 1 many times, it is not clear that this is the case. This claim could be greatly strengthened by showing the movie with another loop that displays the cell tracks. If tracks could be color coded as pre or post phagocytosis, that would help the reader visualize this difference. Further, the data on directional migration is mostly presented as changes in directional persistence (displacement divided by path length), with some additional data on total path lengths mostly restricted in the supplement. However, persistence may not be the most useful metric as these cells are not moving very much. If a cell does not move much or at all, it could still have a very high persistence value. Showing plots with all the tracks (perhaps normalized so that time 0 is at the origin) would be more helpful for understanding if confined and/or post phagocytic cells are really exploring their surroundings more.

Minor Points:

- The images in Figure 3A do not seem representative
- The methods mention the use of a gel loading tip to load the confined cells. It seems this could introduce significant shear forces on the cells. Were the unconfined cells also subjected to this type of pipetting? I doubt this would cause an effect several hours after pipetting, but it may be worth investigating.
- More information in the methods about how the cells were tracked would be helpful (was it based on center of mass?)
- After figure 2, were all experiments with the "post phagocytic" population?
- Line 70, what is a 2.5D environment?
- In figure 4D, did confinement increase persistence in DMSO?
- Why are there fewer Cytochalasin D treated cells in 4H? Is the square trial missing?
- Inverted look up tables could be useful for visualizing the beads
- Lines 98-101 could be rewritten so the second sentence does not seem to contradict the first

Reviewer's Responses to Questions

Experimental quality

Does each figure have the proper controls?

If 'No', please indicate reasons in Comments for Author box below.

Reviewer #1:

- Yes

Reviewer #2:

- No

Were the data analyzed using appropriate statistical tests?

If 'No', please indicate reasons in Comments for Author box below.

Reviewer #1:

- Yes

Reviewer #2:

- Yes

Reproducibility

Were experiments performed using adequate number of biological replicates?

If 'No', please indicate reasons in Comments for Author box below.

Reviewer #1:

- Yes

Reviewer #2:

- No

Does the methods section provide sufficient detail to permit reproducibility?

If 'No', please indicate reasons in Comments for Author box below.

Reviewer #1:

- Yes

Reviewer #2:

- Yes

Completeness

Are the manuscript's conclusions supported by the data?

If 'No', please indicate reasons in Comments for Author box below.

Reviewer #1:

- Yes

Reviewer #2:

- No

Scholarship

Do the authors cite and discuss the merits of data that would argue for and against their conclusion?

If 'No', please indicate reasons in Comments for Author box below.

Reviewer #1:

- Yes

Reviewer #2:

- No

Does the manuscript title & abstract accurately reflect the contents of the manuscript, without hyperbole?

If 'No', please indicate reasons in Comments for Author box below.

Reviewer #1:

- Yes

Reviewer #2:

- No

First revision

Author response to reviewers' comments

We thank both reviewers for their thoughtful comments on our manuscript. We have taken their critiques to heart and have addressed each of their points. We feel the revised manuscript is much stronger because of these reviews.

Reviewer 1: This manuscript from Paulson, Liu, and Rotty describes their efforts to define how physical confinement impacts phagocytosis and motility in microglial cells. The studies use the well-established immortalized mouse line, BV-2. They find that confinement, wherein the BV-2 cells are essentially sandwiched between the underlying substrate and an agarose blanket, promotes phagocytosis of fluorescently labeled IgG. They go on to show, using a collection of validated pharmacological tools, that the Arp2/3 complex and myosin II are required for confinement-mediated phagocytosis. Furthermore, they demonstrate that phagocytosis induces cell motility, which they term "phagocytic priming." Finally, they show that phagocytic priming (cell motility) requires integrin-based adhesion. Overall, this study provides insight into the mechanics of microglial phagocytosis and how phagocytosis promotes cell motility.

The experimental quality is good. The figures, including supplemental, are straightforward to interpret. The assays used are appropriate for addressing the questions asked (e.g., phagocyte formation) and include the proper controls.

Reproducibility is addressed. All experimental data values are plotted along with biological replicate means. At least three replicates are shown for all experiments. The appropriate parametric or non-parametric statistical tests were used based on the data distribution. Necessary details are provided in both the Methods and Figure Legends.

The presented studies are defined and complete. The data largely support the conclusions, though I have suggestions for improvement (mostly on directionality, see below).

Overall, the scholarship is good. The text is clear and well-written. The authors discuss their findings appropriately, citing existing work and incorporating this study into the larger body of work on phagocytosis and cell movement (though I do take issue with one discussion point - see below)

We thank Reviewer 1 for their support and the thorough reading of our manuscript. We have addressed each of the major and minor concerns below.

Major concerns

1. My primary concern is how the authors use persistence and directionality interchangeably. Persistence is not a measure of directionality. Directionality is a bias in one direction (i.e., the movement of a cell toward a chemotactic or haptotactic cue). In contrast, persistence is how long a cell moves in the same direction. The experimental analysis of cell motility used in this study can

provide information on persistence, not directionality. Perhaps I'm missing something here, but the studies do not address haptotaxis or chemotaxis, which could provide insight into directionality. Additional studies are unnecessary, but the authors should be cautious when describing and interpreting the current data.

We completely agree with the reviewer and apologize for this oversight. We have amended the manuscript such that all cell motility is described as persistence and that directionality is not stated or implied. In addition, we have added text to the results section to clarify the difference between directionality and persistence.

2. Along those lines, Figure 1B is misleading as it portrays cell movement in response to a gradient (haptotactic?), but this was not tested/measured.

The figure has been altered to remove the gradient from Figure 1B. This was initially added to indicate microglia moving in the direction of a bead (or beads), but we realize now that this can be construed differently than intended.

Minor concerns

1. Quantifying the phagocytosis experiments required a subjective cutoff to account for variability in bead distribution between substrates. This is okay, but I would like more information on how this was done in the Methods.

The number of beads present in media wells was quantified (Supplemental Figure 1D), and corresponding confined well bead densities were therefore utilized in order to remove bead density as a confounding factor that could impact phagocytosis and motility in one setting more than the other. This has been further clarified in the Methods section as requested.

2. While reading this, I questioned if physical confinement has been shown to regulate persistent movement in other cell types. It turns out it does, as the authors mention in the Discussion. This is an important point and should be raised earlier in the text.

We thank the reviewer for making this suggestion and have added text to the introduction. We have alluded to this in the introduction, while elaborating more specifically on the referenced work in the discussion.

3. I was surprised that blebbistatin treatment did not affect cell movement without beads. Were the authors surprised by this?

We were not initially surprised by this, as there are previous publications that report little, if any, impact of a similar blebbistatin dose on macrophage motility (Rotty et al., *Dev. Cell* 2017; Stinson et al., *MBoC* 2024). However, the effect of blebbistatin can be context-dependent, as macrophages plated on laminin are much more sensitive to blebbistatin than macrophages plated on fibronectin (Stinson et al., *MBoC* 2024). The bead-specific effect of blebbistatin in this manuscript could be a similar context-dependent finding.

4. Could the confinement phenotypes observed with CytoD be caused by differences in uptake (thus altering the effective concentration)? Did the authors try different (higher) concentrations of CytoD? At lower concentrations, CytoD can limit barbed end growth (mimicking capping protein) rather than disrupt the entire actin cytoskeleton.

A dose curve was done in unconfined cells prior to starting the experiment, and concentrations above 1 μ M resulted in too much BV2 cell death to trust quantification outputs. Thus, we believe we are using a high enough concentration of CytoD (1 μ M) to cause significant disruption of the actin cytoskeleton in our cells. We appreciate the reviewer's nuanced view of CytoD, as we have seen similar effects in the past with nM concentrations of the drug. However, the dose that we used would most likely accelerate actin severing as well as limiting barbed end growth. Our expectation is that this would lead to severely compromised actin dynamics.

5. Cilengitide experiments. Perhaps I missed it, but did the authors confirm that

alphaVbeta3 or alpha5beta5 integrins (the targets of Cilengitide) are the primary integrins expressed in BV-5 cells?

We thank the reviewer for raising this point. Previous literature has demonstrated that BV-2 cells do contain $\alpha\text{v}\beta\text{3}$ and $\alpha\text{v}\beta\text{5}$ integrins. These references have been added to the text.

6. The discussion on the MTOC was unwarranted as no data was provided to support a change in MTOC position. I recommend removing this from the Discussion.

We agree upon re-reading that the MTOC section is not as relevant as we initially thought for this manuscript. As such, the discussion of MTOC has been removed from the Discussion section

7. Figure 2 legend. I was confused by the description of how the average phagosome area and average number of phagosomes per cell were calculated. For example, "Average phagosome area per phagocytic cell. This was calculated by dividing the average size of all phagosomes in a field a (of - typo) view by the number of fluorescent cells counted (i.e. phagocytic cells)." Is this correct? Or did you divide the total phagosome area (as measured by fluorescence) in the field by the number of phagocytic cells? These details could also be moved to the Methods.

We thank the reviewer for catching this typo. The average phagosome area per phagocytic cell is the total area of the phagosomes (measured by fluorescence) in the field of view divided by the total number of phagocytic cells in the field of view. We have made the relevant correction to the figure legend.

8. I was surprised that CK-666 did not disrupt persistence as it does in other cell types (like fibroblasts), though it did reduce cell speed (consistent with past data). Might microglia be less dependent on Arp2/3 for cell movement? Should discuss further in the Discussion.

We have added a section to the discussion, as suggested by Reviewer 1. We highlight the fact that the contribution of Arp2/3 complex to persistence seems to be context-dependent. For fibroblasts, FMI (and presumably persistence) is not dramatically affected during chemotaxis, but is significantly lower during haptotaxis. This is also the case with macrophages. Since there is less data on Arp2/3 function in microglia compared to macrophages, we certainly agree with the reviewer that microglia could be much less dependent upon Arp2/3. This could be especially true *in vivo* or in 3D contexts, but it is also a possibility that specific microglial functions would be impacted by Arp2/3 loss.

Reviewer 2: Summary:

The impacts of physical confinement on cell migration have been studied extensively in recent decades, and it is now widely accepted that cells in confinement often migrate differently than cells on coverslips. Despite this, there has been little to no attention paid to how confinement can influence phagocytosis.

This is surprising, given that many of the actin-dependent molecular mechanisms that drive migration also drive phagocytosis. This manuscript by Paulson et al expertly identifies this gap in our collective thinking about cells in physiological contexts. The authors find that confinement increases the rate of phagocytosis, and following phagocytosis cells move with increased directional persistence—a phenomena the authors name "phagocytic priming." Overall, this is an important study that could change the way we think about phagocytosis in physiological environments. While I am overall enthusiastic about this work (and I look forward to sharing this work with my colleagues for a journal club whenever it is public!), I have a few hesitations about some of the methods, conclusions about directional migration, and the statistical analyses.

We thank Reviewer 2 for their overall enthusiasm. We have addressed each of their concerns below. Major Points:

1. Statistical questions: Were the statistical analyses performed on experiment-level averages (N=3-4) or on the entire dataset (dozens to hundreds of cells)? This information is not clear from the methods or legends. If the statistics were done on the entire dataset using the statistical tests as described, I have some concerns about pseudoreplicates artificially deflating the p values. It is

also worth noting if the replicates with the same shape were completed on the same day. Finally, I think it would be useful to show the significance between confined and unconfined cells in DMSO for Fig. 3 B-D, Fig. 4 B-D and F- H, water in 5 B-D, and FN in 5 F-H.

We appreciate the reviewer's concern. Analysis was done on the entire dataset, either the sum total of individual cells (motility) or fields of view (phagocytosis), depending on the graph. This information has been added to the methods section. In response to the reviewer's concerns we have pooled experiments done under similar conditions (e.g. for phagocytosis experiments data from Figs. 2C, 5B 'water', 5F 'FN' and 5E 'water') to achieve N = 13, with a similar pool of DMSO-treated confined and unconfined cells. When plotting the means of these N there is indeed a significant difference between confined and unconfined cells, aside from velocity, for pooled control cells. As our main focus is on phagocytic uptake and directional persistence, these results convince us that our data as a whole is not likely being skewed by pseudoreplicates. We have included these data here for the reviewer, but have not included them in the manuscript as the data is compiled from experiments performed under identical conditions reported throughout the manuscript.

Throughout the paper, shapes were used to denote means of each population within a single experimental run, with circle equating to N = 1, square equating to N = 2, and so forth. Significance between confined and unconfined DMSO values in each of our drug response graphs has been added (and also the compiled DMSO +/- confinement mean values are included here, as stated above).

2. Were the unconfined cells in complete media (as described in the Figure 1 legend, line 521), or PBS (as described in methods, line 393)? Either way, the agarose solution appears to be a bit different (HBSS and media as described in methods). Because these solutions are different, is it possible that the different osmolarities, salt concentrations, pHs, etc. could be driving the difference between unconfined and confined cells?

We thank the reviewer for pointing out this possibility. Unconfined cells were in complete media for all experiments as described in the Figure 1 legend and the results section. The PBS described in line 393 of the original manuscript was used when initially creating the dishes and storing them long term.

Description detailing replacing the PBS with complete media in preparation for cell seeding has been added to the methods. We thank the reviewer for the perceptive point about osmolarity, salt, and pH potentially driving the differences. However, the composition of HBSS has been engineered

to maintain physiological osmolarity and pH, and is at a 1X final concentration in the gel. The recipe we use for making the gels is nearly identical to the recipe used by Heit and Kubes (2003, *STKE*), except that we use DMEM + FBS instead of RPMI + FCS. The HBSS-containing gels are the standard in the field, but we agree with the reviewer that in principle the different composition could be driving the changes we see in the confined setting. In order to rule this out, we include data in the revision showing that unconfined cells cultured in DMEM + 1X HBSS (identical to agarose gel composition, minus the agarose) are still less phagocytic than cells confined in the HBSS-containing gel. These data strongly support physical confinement as the factor that drives the enhanced phagocytic uptake phenotype. These data can be found in the new supplemental figure S2C.

3. How confined are the cells? They do not look particularly flattened in movie 1, and the beads seem to be floating/flowing, suggesting the ceiling is higher than 2 microns. Is it possible to measure the height?

We thank the reviewer for bringing up this important point. To address it, we made agarose gels with fluorescent dextran, or added dextran to media, and imaged cells via confocal after beads were washed in. Based on the z-step size we were able to quantify a rough estimate of cell height. Unsurprisingly, unconfined cells are fairly tall, compared to confined cells which are much more compact. Cells that had taken up beads were still confined by the agarose, but managed to lift the agarose ceiling somewhat. This is perhaps due to the rigidity of the engulfed latex beads, or the stiffness of the cell body though we do not speculate about this in the manuscript. Z-projection views illustrate that confined cells are tightly surrounded by agarose. Beads injected under the agarose will by definition lift the gel while beads are introduced, due to fluid flows. However, the agarose gel seems to settle back to the cells/coverlip without incident. There may be some regions where this takes longer or occurs imperfectly, but for the most part we feel that these data confirm that cells are still tightly confined for most of the time that they are interacting with beads. These data are in new supplementary figure S2A-B, and a description has been added to the methods.

4. Is it possible the heat from the agarose inactivated some of the drugs? This could explain how an agarose pad rescued the Cytochalasin D phenotype, and would also explain the lamellipodia-like structures that seemed to be abundant in video S2 in the presence of CK-666. Some type of control to prove the drugs are still effective would be useful (some ideas: removing the agarose pad if possible? Having cells crawl on top of the agarose pad instead? Making a gel with drugs at 2x, incubating in an equal volume of drug-free media, then testing that media on unconfined cells?).

We thank the reviewer for pointing this out as a confounding factor. In our hands it is not possible to remove the agarose pad without disrupting the cells. We considered placing the BV-2 cells on top of the agarose, but this seemed too challenging logistically both from an imaging perspective and a cell health perspective. Instead, we attempted the reviewer's last suggestion and have done this experiment, incubating media with a 1 μ M cytoD containing gel, and then testing the media on unconfined cells to determine how much cytochalasin D has leached out of the gel. We added a 1:1 ratio of media and gel, so if 1 μ M cytoD activity is well-maintained we should expect to perform similarly to a 0.5 μ M dose. We found that the gel-derived cytochalasin D still robustly affected unconfined cells' phagocytic ability similarly to a 0.5-1x (0.5-1 μ M) dose added directly to unconfined cells. Thus, we feel confident that the drug dose applied to the gel is quite close to what we expect it to be, and that heat inactivation is likely not playing a significant role. We have added these data in Supplemental Figure S4E.

5. Conclusions about directed migration. A central point of the manuscript (and the title) is that phagocytosis increases directional migration. For example, line 149 claims that "bead uptake seemed to induce more persistent migration tracks" with a reference to Movie 1. From watching movie 1 many times, it is not clear that this is the case. This claim could be greatly strengthened by showing the movie with another loop that displays the cell tracks. If tracks could be color coded as pre or post phagocytosis, that would help the reader visualize this difference. Further, the data on directional migration is mostly presented as changes in directional persistence (displacement divided by path length), with some additional data on total path lengths mostly restricted in the supplement. However, persistence may not be the most useful metric as these cells are not moving

very much. If a cell does not move much or at all, it could still have a very high persistence value. Showing plots with all the tracks (perhaps normalized so that time 0 is at the origin) would be more helpful for understanding if confined and/or post phagocytic cells are really exploring their surroundings more.

We thank the reviewer for pointing out this oversight. We have included plots now demonstrating the tracks (with time 0 as the origin) in Figure 2J. We have also included example movies (with track overlays) that show pre- and post-phagocytic cell migration (Supplemental Movie 2).

Minor Points:

- The images in Figure 3A do not seem representative

New representative images have been selected and substituted.

- The methods mention the use of a gel loading tip to load the confined cells. It seems this could introduce significant shear forces on the cells. Were the unconfined cells also subjected to this type of pipetting? I doubt this would cause an effect several hours after pipetting, but it may be worth investigating.

We thank the reviewer again for bringing this potential confounding factor to our attention. While shear forces could be a factor, we do not think it influences phagocytic priming, especially as unconfined cells still demonstrate phagocytic priming ability. As phagocytosis occurs at a consistent rate throughout the time lapse, we do not think acute shear forces are playing a role in this persistent cellular behavior.

- More information in the methods about how the cells were tracked would be helpful (was it based on center of mass?)

Cells were tracked based on the base of the forward-most protrusion. Text defining this has been added to the methods section.

- After figure 2, were all experiments with the "post phagocytic" population?

Yes, this is correct. After Figure 2, all experimental data referring to cell motility was with the post- phagocytic population. We will add text stating this to all of the relevant figure legends and to the methods.

- Line 70, what is a 2.5D environment?

This line, referencing a study by Sharaf et al., CAD printed nanopillars at equal spacing and placed microglia across them, referring to this as a 2.5D environment, as it had no 3rd dimension, but was different from cells plated on a flat surface. This study also CAD printed cubes made out of nanopillars, that they referred to as 3D and differed from their 2.5D studies. We have added text to clarify what we mean by 2.5D.

- In figure 4D, did confinement increase persistence in DMSO?

Confinement did not increase persistence in a significant fashion in the presence of DMSO in this experiment.

- Why are there fewer Cytochalasin D treated cells in 4H? Is the square trial missing?

This is due to the fact that there were many fewer phagocytic cells under this condition. As the motility measures were reported post-phagocytosis, there are just naturally fewer cells that fit the inclusion criteria for this analysis. We have added text to the relevant results and methods section to make this clearer.

- Inverted look up tables could be useful for visualizing the beads

We thank the reviewer for this suggestion, but think that the current LUT with red beads is sufficient to demonstrate the difference between internalized and uninternalized beads.

- Lines 98-101 could be rewritten so the second sentence does not seem to contradict the first

We thank the reviewer for this feedback. Lines 98-101 have been rewritten to not come across as contradictory.

Second decision letter

MS ID#: bio.062021R1

MS Title: Physical confinement and phagocytic uptake induce persistent cell migration

Authors: Summer G. Paulson, Sophia Liu and Jeremy Rotty

I have now reached a decision on the above manuscript.

The reviewer reports are shown at the bottom of this email or can be accessed, together with a copy of this decision letter, by going to:

Both reviewers are positive, as you will see, but reviewer 2 has requested some very minor change. As no further changes to the manuscript are possible after acceptance, I want to give you the opportunity to address these comments. I hope that you will be able to carry these out, because we would like to be able to accept your paper.

At this stage, we also ask you to ensure your manuscript complies with our formatting guidelines “please see our manuscript preparation guidelines for details. Provided you are able to fully address the referees’ comments, we are positive about publication of your paper (we accept over 95% of revision submissions) and therefore hope you won’t mind any extra work involved in reformatting your manuscript at this point.

Please ensure that you clearly highlight all changes made in the revised manuscript. Please avoid using 'Tracked changes' in Word files as these are lost in PDF conversion.

I should be grateful if you would also provide a point-by-point response detailing how you have dealt with the points raised by the reviewers in the 'Response to Reviewers' box. Please attend to all of the reviewers’ comments. If you do not agree with any of their criticisms or suggestions please explain clearly why this is so.

Reviewer 1

Comments for the author

The authors have addressed all my primary concerns. The initial version of the manuscript was well-written and described a robust set of experiments, and the revised version is significantly improved. The work provides new and important information into the how phagocytosis can regulate cell movement.

Reviewer 2

Comments for the author

Thank you for completing the requested experiments and analyses, and for responding to all of my concerns. I have only 2 very minor suggestions:

1. The figure legends should be explicit that stats were performed on "n" and not "N."
 2. Thank you for including the HBSS experiment. While I think this is important to include in the manuscript, it may be more appropriate to move the discussion of this (starting on line 169) to the methods section, as it is breaking up the flow of your writing.
-

Second revision

Author response to reviewers' comments

Reviewer 1: The authors have addressed all my primary concerns. The initial version of the manuscript was well-written and described a robust set of experiments, and the revised version is significantly improved. The work provides new and important information into the how phagocytosis can regulate cell movement.

We thank Reviewer 1 for the helpful comments, which substantially improved the final version of the manuscript.

Reviewer 2: Thank you for completing the requested experiments and analyses, and for responding to all of my concerns. I have only 2 very minor suggestions:

1. The figure legends should be explicit that stats were performed on "n" and not "N."
Figure legends and Supplemental Figure legends have been updated to explicitly state that stats were performed on the n and not the N.

2. Thank you for including the HBSS experiment. While I think this is important to include in the manuscript, it may be more appropriate to move the discussion of this (starting on line 169) to the methods section, as it is breaking up the flow of your writing.

We thank the reviewer for this feedback and agree. The Supplemental Figure reference has been moved earlier in the paragraph. The HBSS section starting at line 169 has been removed, and more detail on the HBSS experiment can be found in the methods section starting at line 575.

We thank Reviewer 2 for their continued comments, which also substantially improved the final version of the manuscript.

Third decision letter

MS ID#: bio.062021R2

MS Title: Physical confinement and phagocytic uptake induce persistent cell migration

Authors: Summer G. Paulson, Sophia Liu and Jeremy Rotty

I am happy to tell you that your manuscript has been accepted for publication in Biology Open, pending our standard publication integrity checks. It was accepted on 13th August 2025. I appreciate very much your quick response, and attention to the final comments of Reviewer 2.